# Best Fit 3D Basilar Membrane Reconstruction to Routinely Assess the Scalar Position of the Electrode Array after Cochlear Implantation

**DOI:** 10.3390/jcm11082075

**Published:** 2022-04-07

**Authors:** Renato Torres, Jean-Yves Tinevez, Hannah Daoudi, Ghizlene Lahlou, Neil Grislain, Eugénie Breil, Olivier Sterkers, Isabelle Mosnier, Yann Nguyen, Evelyne Ferrary

**Affiliations:** 1Technologies et Thérapie Génique Pour la Surdité, Institut de l’Audition, Inserm/Institut Pasteur/Université de Paris, 75012 Paris, France; hannah.daoudi@aphp.fr (H.D.); ghizlene.lahlou@aphp.fr (G.L.); o.sterkers@gmail.com (O.S.); isabelle.mosnier@aphp.fr (I.M.); yann.nguyen@inserm.fr (Y.N.); evelyne.ferrary@inserm.fr (E.F.); 2Unité Fonctionnelle Implants Auditifs et Explorations Fonctionnelles, Service ORL, GHU Pitié-Salpêtrière, Assistance Publique-Hôpitaux de Paris (AP-HP)/Sorbonne Université, 75013 Paris, France; neil.grislain@aphp.fr (N.G.); eugenie.breil@gmail.com (E.B.); 3Departamento de Ciencias Fisiológicas, Facultad de Medicina, Universidad Nacional de San Agustín de Arequipa, Arequipa 04002, Peru; 4Image Analysis Hub, Institut Pasteur, Université de Paris, 75015 Paris, France; jean-yves.tinevez@pasteur.fr

**Keywords:** hearing loss, hearing impairment rehabilitation, scala vestibuli, scala tympani, auditory prosthesis, electrode array translocation

## Abstract

The scalar position of the electrode array is assumed to be associated with auditory performance after cochlear implantation. We propose a new method that can be routinely applied in clinical practice to assess the position of an electrode array. Ten basilar membrane templates were generated using micro-computed tomography (micro-CT), based on the dimensions of 100 cochleae. Five surgeons were blinded to determine the position of the electrode array in 30 cadaveric cochleae. The procedure consisted of selecting the appropriate template based on cochlear dimensions, merging the electrode array reconstruction with the template using four landmarks, determining the position of the array according to the template position, and comparing the results obtained to histology data. The time taken to analyze each implanted cochlea was approximately 12 min. We found that, according to histology, surgeons were in almost perfect agreement when determining an electrode translocated to the scala vestibuli with the perimodiolar MidScala array (Fleiss’ kappa (κ) = 0.82), and in moderate agreement when using the lateral wall EVO array (κ = 0.42). Our data indicate that an adapted basilar membrane template can be used as a rapid and reproducible method to assess the position of the electrode array after cochlear implantation.

## 1. Introduction

Cochlear implantation is a surgical procedure to insert an electrode array into the cochlea. This device stimulates the ganglion auditory cells and rehabilitates hearing; however, postoperative auditory performance can vary from patient to patient. Many preoperative factors, such as the etiology of hearing loss, duration of profound deafness, use of hearing aids, and age at implantation [1,2] can affect the hearing outcome; however, the electrode array insertion is one of the few factors that can be optimized. During cochlear implantation, it is important to accomplish non-traumatic insertion of the electrode array, as this is associated with a reduction in inflammatory processes [3], preservation of residual hearing [4], and improvement in hearing performance [5,6,7].

Different surgical strategies have been developed to improve electrode array insertion, such as using fine and flexible electrode arrays [6], round window insertion [8], and using hyaluronic acid after opening the cochlea to lubricate it and avoid blood contamination and perilymph leakage [9]. Most recently, robot-assisted insertion of the electrode array has been implemented to improve the accuracy of movement during surgery [10]. Despite the development of these technical and surgical strategies, there have been a number of reports of incorrect location of the electrode array, with varying degrees of translocation from the scala tympani to the scala vestibuli [5,6,11,12,13]. It is therefore essential to determine whether an electrode array is poorly positioned after surgery, as this will result in poor hearing performance, requiring a technical adjustment of the cochlear implant processor.

Different methods have been proposed to precisely determine the position of the electrode array and its possible translocation. Computed tomography (CT) is regularly performed after cochlear implantation to determine the presence of any translocation. Earlier studies have reported the use of rotational tomography [14], multisectional CT [15], and cone-beam CT imaging (CBCT) [16,17]. Although different multiplanar reconstructions have been performed to assess the position of electrode arrays, they have major limitations owing to the blurring effect produced by the metallic artifacts of the electrodes. Because intracochlear structures are not visible with post-implantation imaging, another reported method is the fusion of postoperative CT imaging with preoperative magnetic resonance imaging (MRI) [18,19]. MRI is routinely performed before cochlear implantation, and can be merged with postoperative imaging. However, the number of slices passing through the cochlea is limited, and to improve image quality, the time of acquisition must be extended significantly, which is not practical in a clinical scenario. To overcome the variability of cochlear anatomy, a method has been proposed that uses manual 3D reconstruction of the basilar membrane on preimplantation CBCT [20]. This 3D reconstruction was merged with the 3D reconstruction of the electrode array from the same patient based on the position of the semicircular canals. Although the accuracy of this method has been validated using histology, the time required to obtain a 3D reconstruction of the basilar membrane is a limitation.

Because intracochlear structures are not visible on postoperative imaging, 3D reconstruction models have been used to determine the position of the electrode array after cochlear implantation. A rigid model was proposed as a method for estimating the position of intracochlear structures [21,22]. The high-quality rigid model obtained was then merged with the CT images to determine the position of the electrode array. However, with a rigid model, the variability of the cochlear anatomy parameters, such as its dimensions [23], or the variability of the coiling of the cochlea [24,25], cannot be taken into account. To improve the accuracy of the method, the use of several rigid cochlear models would be a way to adapt to anatomic variations. Another proposed method used nonrigid models to determine the position of the electrode array [26,27]. This allowed the model to be adjusted according to the cochlear anatomy, and automatically determined the position of the intracochlear structure. However, 3D reconstruction methods are time-consuming, require considerable manual effort to handle images, and require the necessary knowledge and training; thus, they are not included in the clinical procedure. Consequently, a method that allows for routine and rapid analysis of the position of the electrode array in clinical practice is still lacking, but desirable.

In this study, we evaluated the accuracy of determining the position of the electrode array using a 3D basilar membrane template selected to match the cochlear proportions of 30 cadaverically implanted cochleae. The position of each electrode was assessed by five ear-nose-throat (ENT) surgeons and compared with the histology.

## 2. Materials and Methods

### 2.1. Cochlear Images Used in This Study

We based this study on an image bank that included the following:A set of 100 CT images from pre-implanted patients;Images of 30 cadaveric temporal bones, including pre- and post-implantation CBCT;A collection of 22 CT images of non-implanted cochleae—20 from the SMIR database of cochlea data descriptors (SICAS Medical Repository, Corroux, Switzerland) [28] and 2 from our own database.

### 2.2. Registration Procedure

#### 2.2.1. Determination of Cochlea Dimensions

Selection of the “basilar membrane” template for each patient was based on several measurements of the cochlea imaged by preoperative CT. These measurements were taken in a single slice at a specific orientation and position through the cochlea. In the first implementation of the method, we used the 3D multiplanar reconstruction viewer of the General Public License (GPL) software Horos v.3.3.6 (Horos project, Geneva, Switzerland) [29]. The cochlea was aligned as follows: (1) the intersection of the three planes was placed on the mid-modiolar axis; (2) the coronal plane was aligned with the middle plane of the basal turn (center of the round window, the middle of the cochlear turn at 90°, 180°, and 270°); (3) the axial plane was consequently perpendicular to the coronal plane and passed through the center of the round window and the cochlear turn at 180°; and (4) the sagittal plane was also perpendicular to the coronal plane and passed through the cochlear turn at 90° and 270°.

Aligning the planes with the theoretical position of the middle plane of the cochlear turn is crucial. In a post-implanted cochlea, the theoretical position of this plane is defined without considering the position of the electrode array. Once alignment was completed, we measured the following (Figure 1):

Distance A (A) between the center of the round window and lateral wall at 180°;Distance B (B) between the lateral wall at 90° and 270°;The height at 360° (H^360^), measured from the base of the cochlea to the highest point of the cochlear turn at 360°.

We defined three cochlear indices from these measurements (Table 1): A×H^360^, (A×B) ×H^360^, and (A/B)×H^360^.

#### 2.2.2. Determination of the Position of Four Electrode Array Landmarks

On the same slice, we then determined the 3D coordinates (x, y, z) of the four landmarks of the electrode array (Figure 2) that characterize its extent in 3D: the center of the round window, and the lateral wall at 90°, 180°, and 270°.

#### 2.2.3. “Basilar Membrane” Segmentation

Where appropriate, the middle plane of the cochlear turn was manually segmented from the non-implanted cochleae images to obtain a 3D reconstruction, as reported in previous studies [13,20], using the GPL software ITK-SNAP v.3.4.0 (U.S. National Institutes of Health) [30]. Reconstruction of the middle plane of the cochlear turn included segmentation of the spiral lamina and basilar membrane. Throughout this paper, the term “basilar membrane” reconstruction represents reconstruction of the middle plane of the cochlear duct.

#### 2.2.4. Electrode Array Segmentation

The electrode array was automatically segmented from CT images of the implanted cochlea. Because of its metallic composition, the array appeared as a very bright structure with pixel values well above those of the pixels in the temporal bone. We automatically segmented the electrode array volume using a threshold, as previously reported [13,20].

#### 2.2.5. Procedure for Merging 3D Reconstruction Models

Finally, the two 3D reconstructions (“basilar membrane”/electrode array or “basilar membrane”/”basilar membrane”) were registered and merged in the same scene using the four corresponding landmark points in each reconstruction (center of the round window, lateral wall at 90°, 180°, and 270°), with the point-by-point tool of the GPL software CloudCompare v.2.10.2 (https://www.cloudcompare.org) [31].

### 2.3. The “Basilar Membrane” Templates

Here, we describe how we built the five “basilar membrane” templates used after registration to determine the scalar position of the array. These five templates were built such that one could represent the cochlea of any patient once properly scaled. We also derived a procedure to select the best template based on the patient’s cochlear dimensions as follows:

First, the cochlear dimensions and the three cochlear indices were obtained from pre-implantation CT images from 100 patients. The values obtained followed a Gaussian distribution ((A)×H^360^: *p* = 0.82; (A×B)×H^360^: *p* = 0.85; (A/B)×H: *p* = 0.13; Shapiro–Wilk test). Consequently, five micro-CT cochleae were selected according to the Gaussian distribution of the patients’ CT imaging (Figure 3), and this procedure was repeated for the three indices. In each case, the “basilar membrane” reconstruction and the four landmarks were obtained.

Second, we reconstructed 10 “basilar membranes” from CBCT images of cadaveric temporal bones. The cochlear index was calculated based on each CBCT image, and the corresponding “basilar membrane” template was selected. Then, both “basilar membranes” were merged by means of the corresponding landmarks using the point-to-point tool in CloudCompare. Because it is quite difficult to manually segment the basilar membrane in CBCT images (hook region and beyond 540°), both “basilar membranes” (micro-CT and CBCT) were cropped from 90° to 540° at some locations. 

Finally, the mean distances between the reconstruction models were calculated using the cloud/cloud distance tool of CloudCompare (Figure 4). 

This procedure was repeated for each CBCT image and for the three cochlear indices. The distance after merging both “basilar membranes” was similar using the three indices ((A)×H^360^: 0.15 ± 0.03; (A×B)×H^360^: 0.16 ± 0.02; (A/B)×H^360^: 0.14 ± 0.02; *p* = 0.08, Kruskal–Wallis and Bonferroni post hoc test). The (A/B)×H^360^ index was considered to be the best fit index because of its smaller distance error. We then used the GPL software Blender 2.8.0 (Blender Foundation, Amsterdam, The Netherlands) [32] to generate the five contralateral “basilar membrane” reconstructions.

### 2.4. Determination of the Intrascalar Position of Each Electrode

The best “basilar membrane” template was selected based on the (A/B)×H^360^ index measured on pre-implantation CBCT. The electrode array was segmented from the post-implantation CBCT and merged with the “basilar membrane” template using the four corresponding 3D points.

The scalar position of each electrode was determined according to the “basilar membrane”, as the scala tympani, intermediary, and scala vestibuli electrodes. Owing to the rigidity of the “basilar membrane” reconstruction, the position of the electrode array was defined as follows:Scala tympani electrode: ≥50% of the electrode under the “basilar membrane”.Intermediate electrode: ≥10% to <50% of the electrode under the “basilar membrane”.Scala vestibuli electrode: <10% under the “basilar membrane”.

### 2.5. Comparing the “Basilar Membrane” Reconstruction with Histology to Determine the Intrascalar Position of the Electrode Array

An ENT surgeon who did not participate in the image analysis chose 30 implanted cochleae from our histopathologic database (7 cochleae with translocations and 8 without for the HiFocus™ Mid-Scala electrode array (Advanced Bionics, Valencia, CA, USA); and 8 cochleae with translocations and 7 without for the Digisonic® SP EVO electrode array (Oticon Medical, Vallauris, France). The AB MidScala electrode array has 16 electrodes and the Oticon EVO electrode array has 20 electrodes. Each cochlea had a pre- and post-implantation CBCT. The same surgeon prepared a file with the pre- and post-operative CBCT for the 10 “basilar membrane” reconstructions corresponding to the (A/B)×H^360^ index, and gave it to the five surgeons for analysis. Any information on the histopathological study was made available to the surgeons. Two ENT surgeons were considered experts because of their experience in handling 3D reconstruction models. The other three ENT surgeons were considered non-experts because they had never handled 3D reconstruction models before this study.

Each surgeon performed the following procedure for each case:Select a “basilar membrane” template according to the index value obtained on pre-implantation CBCT.Obtain the reconstruction of the electrode array from the post-implantation CBCT.Obtain the four corresponding points from the post-implantation CBCT.Merge the electrode array reconstruction with the selected “basilar membrane” template according to the four landmarks.Determine the position of each electrode.

Finally, the positions of each electrode determined by the described technique were compared with the histopathological analysis, which served as the ground-truth reference (Figure 5).

### 2.6. Description of the Histopathological Analysis

All cochleae used in this study were analyzed and the results stored in our database. After electrode array insertion, the cochlea was harvested, and the apex of the cochlea and lateral semicircular canal were opened. The cochlea was then fixed with 10% formaldehyde for 24 h, dehydrated with increasing concentrations of alcohol (from 50% to 100%) for 12 h, dried at ambient air temperature for 16 h, and fixed with a crystal resin (Pebeo, Gémenos, France) until polymerization, as reported in a previous study [20]. The cochlea was then progressively ground perpendicular to the round window/modiolar axis and stopped at the level of each electrode. The cochlea was stained with Phloxine B for 15 min and visualized with a stereomicroscope (SLM 2; Karl Kaps GmbH, Wetzlar, Germany). A photograph was taken and stored in our database.

### 2.7. Statistical Analysis

The cochlear dimensions and index values are expressed as mean ± standard deviation. The normal distribution of the cochlear dimensions was checked using the Shapiro–Wilk test. The Fleiss’ kappa coefficient was used to analyze the agreement between determining the scalar position of each electrode using a “basilar membrane” template and the histopathological study, in relation to the surgeons’ experience in handling 3D reconstructions and the associated software. Data were analyzed using R statistical software v3.3.3 (R Core Team, Vienna, Austria).

## 3. Results

### 3.1. Inter-Rater Agreement in Determining the Scalar Position of Each Electrode

The entire procedure for determining the scalar position of the electrode took less than 12 min in all cases. We observed substantial agreement in determining the scalar position of each electrode of the AB MidScala electrode array, regardless of the experience of the surgeon (κ = 0.68 (0.66–0.71)). There was also substantial agreement between expert and non-expert surgeons in determining the scalar position of each electrode (κ = 0.76 (0.71–0.83) and κ = 0.67 (0.62–0.71), respectively). For the Oticon EVO electrode array, the agreement among surgeons regardless of their experience was κ = 0.39 (0.37–0.41). Experts had better agreement than non-experts in determining the scalar position of each electrode (κ = 0.46 (0.41–0.51) and κ = 0.39 (0.35–0.42), respectively). The overall agreement in determining the position of each electrode was decreased because of the higher number of electrodes in an intermediate position with the AB MidScala array (13/250 electrodes assessed by histology) than with the Oticon EVO array (50/300 electrodes) (*p* < 0.001; chi-squared test). With regard to the electrodes in the scala vestibuli, the agreement in determining the scalar position of each electrode was almost perfect for the MidScala array, regardless of the experience of the surgeon, and moderate for the Oticon EVO array (κ = 0.82 (0.79–0.84) and κ = 0.42 (0.40–0.45), respectively), (Table 2).

### 3.2. Different Translocation Patterns Depending on the Type of Electrode Array

Different translocation patterns were observed depending on the type of electrode array (perimodiolar or lateral wall). With regard to the position of the array translocation, the MidScala electrode array translocated around 180° and rapidly crossed the middle plane of the cochlear duct, but beyond this point, the array remained in the scala vestibuli. Distal translocation of the array was not observed with this electrode array. In contrast, the Oticon EVO array was translocated around 180° (proximal translocations) and/or beyond 300° (distal translocations). As observed in Figure 6, the electrode array bent upward into the cochlear duct, pushing the basilar membrane, and the array remained in an intermediate position on the lateral wall.

## 4. Discussion

In this study, we proposed and evaluated a rapid and reproducible technique to precisely assess the position of the electrode array according to the middle plane of the cochlear duct. To achieve this, a “basilar membrane” template was selected based on cochlear dimensions, and merged with the electrode array reconstruction to determine the position of the electrode array. In comparison with histology data, there was almost perfect agreement in determining an electrode translocated to the scala vestibuli with the AB MidScala array and a moderate agreement with the Oticon EVO array.

We selected five cochleae from a micro-CT database based on the best index obtained with the cochlear dimensions: horizontal dimensions, distances A and B; and vertical, H^360^. Regarding the selection of H^360^ as a metric to select the “basilar membrane” template, previous reports have shown that the number of turns in the population is variable (from less than 2.5 to 3 turns) [33]. This could influence the slope of the middle plane of the cochlear turn, and two cochleae with the same full height but different numbers of turns would have marked differences in basilar membrane position. Indeed, the basilar membrane slope is higher in a cochlea with fewer turns than in one with more turns. In addition, in exceptional circumstances, the electrode array insertion depth can be greater than 540° for all electrode array types; thus, the region of interest to detect the position of the array would range from 0° to 450° [6,12,20,34,35]. Consequently, the height of the cochlea at 360° is a more robust selection criterion for the “basilar membrane” template.

Earlier studies reported different methods to assess the position of the electrode array after cochlear implantation, such as the multiplanar reconstruction of postoperative imaging [14,15,16], merging pre- and post-operative imaging [18,19], rigid models [21,22], manual segmentation of the basilar membrane [20], and non-rigid models [26,27]. Each of these techniques has been used to determine the position of the electrode array; however, because of the cognitive load and time required for analysis, they are not widely included in the clinical workflow. A fundamental advantage of the proposed technique is the reduction in the time required for analysis. In addition, this method requires some measurements to be regularly performed by a surgeon, and the visual aspect offered by 3D models could provide a global insight into array insertion, as well as into the determination of the scalar position of each electrode. However, a limitation of this technique is that it uses several software tools for the analysis. Future efforts will focus on producing an all-encompassing and specialized software tool to support this method, which would further diminish the time and effort required. Furthermore, a fully automated procedure for determining the position of each electrode is achievable. This automation could mitigate the inter-rater variability and facilitate its dissemination in the clinical context.

Our data indicate a different translocation mechanism depending on the model of inserted electrode array. For the precurved AB MidScala array, translocations were observed at approximately 180°, as confirmed by other reports [13,36]. Owing to the lateral rigidity of this array, a misalignment with the direction of the scala tympani would lead to perforation of the basilar membrane and subsequent translocation, as reported in an earlier study [37]. The straight Oticon EVO array would progress in contact with the lateral wall, and the frictional forces between the array and the lateral wall would progressively increase. Consequently, the array could bend inside the scala tympani, push the basilar membrane upward, and produce detachment of the basilar membrane.

In summary, we have established a rapid and reproducible method using a 3D “basilar membrane” template selected to match the dimensions of the cochlea, which enables the scalar position of each electrode to be detected after cochlear implantation. Further studies are required to automate the entire procedure before its introduction into routine use in clinical practice.

## Figures and Tables

**Figure 1 jcm-11-02075-f001:**
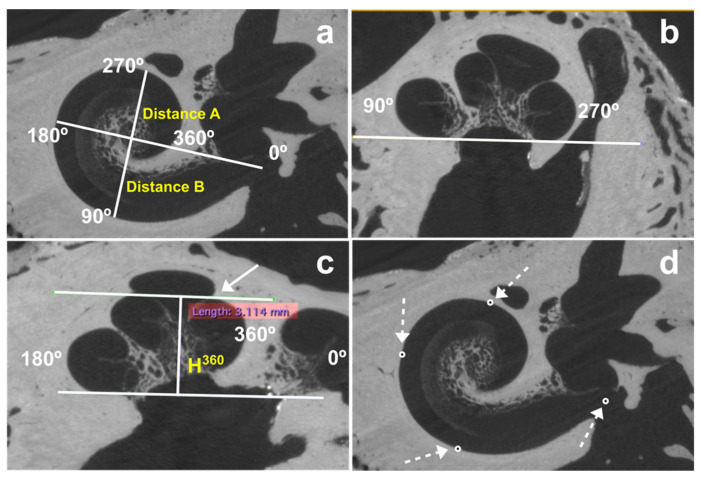
Measurement of distances A, B, and H^360^. H^360^ was measured from the base of the cochlea to the highest part of the cochlear turn at 360° (white arrow). (**a**) Coronal view, (**b**) sagittal view, which has been rotated to position the basal turn inferiorly for better visualization, (**c**) axial view. (**d**) the same alignment was necessary to measure distances A and B to determine the position of the center of the round window and the lateral wall at 90°, 180°, and 270° (white discontinuous arrows).

**Figure 2 jcm-11-02075-f002:**
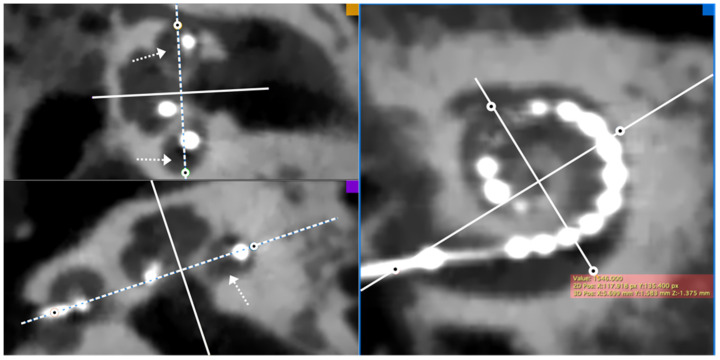
Determination of the 3D positions of four landmarks (the center of the round window, and the lateral wall at 90°, 180°, and 270°) on a post-implantation CBCT image. Note that the coronal plane (dashed line) is aligned with the middle of the cochlear turn regardless of the position of the electrode (white arrows). The point corresponding to the intersection between the middle cochlear turn line and the lateral wall was selected.

**Figure 3 jcm-11-02075-f003:**
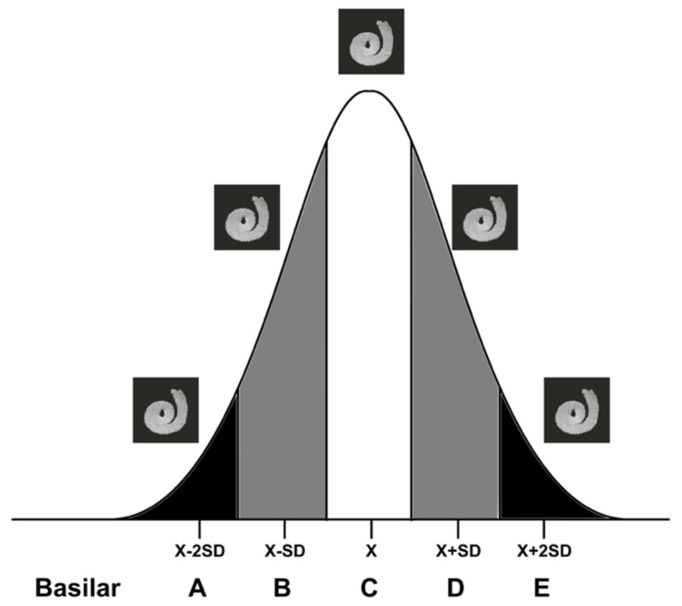
Five “basilar membrane” templates selected according to the normal distribution of each index—(A)×H^360^, (A×B)×H^360^, and (A/B)×H^360^—on 100 CT images from pre-implanted patients.

**Figure 4 jcm-11-02075-f004:**
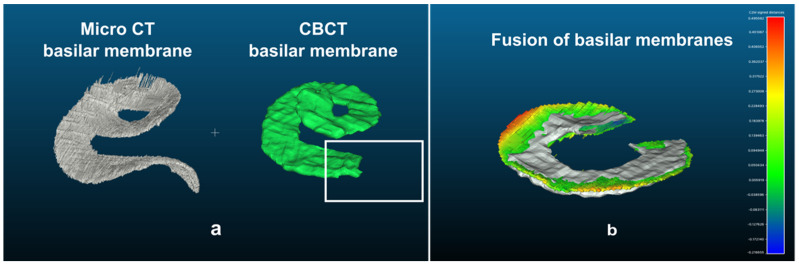
Comparison of basilar reconstruction segmented from micro-CT images and CBCT. The term basilar membrane represents the middle plane of the cochlear turn which includes the basilar membrane and the spiral lamina. (**a**) The hook region has not been segmented on CBCT images because of its complexity (open square); (**b**) the distance between both segmented “basilar membranes” (both cropped from 90° to 540°) was calculated. The color scale represents the distance between both basilar membranes, from green (shorter gap) to red (greater gap). In grey, both basilar membranes are superposed.

**Figure 5 jcm-11-02075-f005:**
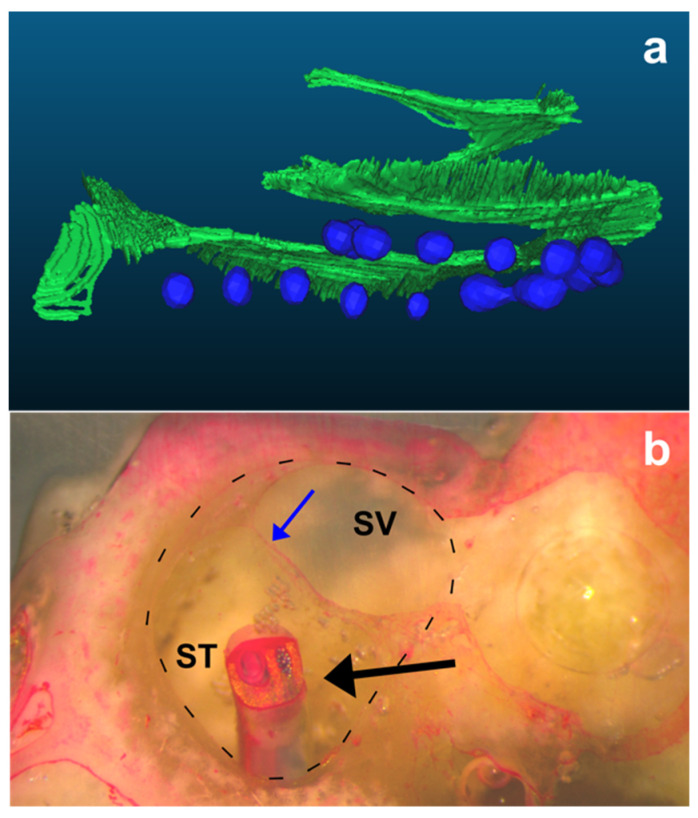
The middle plane of the cochlear turn reconstruction and histologic techniques were used to assess the position of each electrode array. (**a**) The “basilar membrane” reconstruction, including the basilar membrane and spiral lamina, was selected based on the dimensions of the cochlea analyzed; (**b**) the microgrinding technique shows the electrode array penetrating the scala tympani. The black discontinuous lines delimit the cochlear duct. ST: scala tympani; SV: scala vestibuli; black arrow: electrode array; blue arrow: basilar membrane.

**Figure 6 jcm-11-02075-f006:**
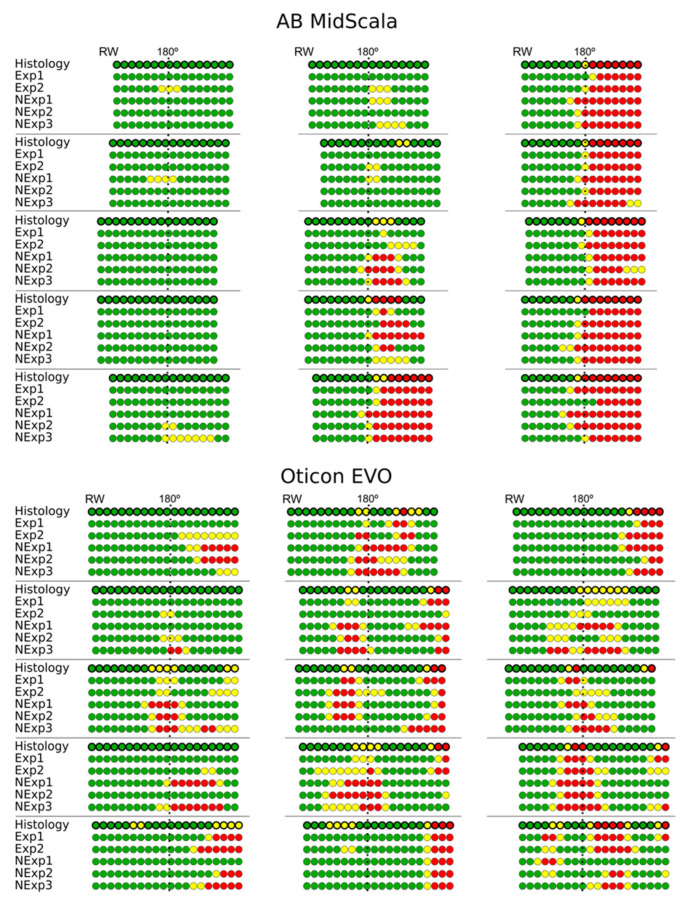
Assessment of the scalar position of the electrodes by five raters according to histology. The electrode array was located according to the positions of the round window (RW) and at 180° (dotted lines). Exp: expert; NExp: non-expert; scala tympani electrode: green circle; intermediate electrode: yellow circle; scala vestibuli electrode: red circle.

**Table 1 jcm-11-02075-t001:** Cochlear dimensions of 100 pre-implantation cochleae. The three indices (A)×H^360^, (A×B) ×H^360^, and (A/B)×H^360^ have a Gaussian distribution.

	Patient CT (*n* = 100)		Cadaveric CBCT (*n* = 30)		Micro-CT (*n* = 22)	
	Mean ± SD	Min–Max	Mean ± SD	Min–Max	Mean ± SD	Min–Max
Distance A	9.1 ± 0.30	8.0–9.6	9.1 ± 0.22	8.7–9.4	9.2 ± 0.33	8.6–9.8
Distance B	6.8 ± 0.32	5.8–7.6	6.9 ± 0.24	6.6–7.4	7.0 ± 0.31	6.5–7.5
H^360^	2.8 ± 0.21	2.4–3.3	2.9 ± 0.17	2.6–3.3	2.9 ± 0.19	2.3–3.3
(A)×H^360^	26 ± 2.30	20–33	26 ± 1.9	23–30	27 ± 2.4	22–29
(A×B)×H^360^	175 ± 19.9	126–210	182 ± 17.2	159–211	194 ± 23.3	135–203
(A/B)×H^360^	3.8 ± 0.30	3.0–4.7	3.8 ± 0.24	3.4–4.2	3.9 ± 0.29	3.1–4.4

CT: computed tomography; CBCT, cone-beam CT.

**Table 2 jcm-11-02075-t002:** Agreement of the five surgeons (two experts and three non-experts) in determining the position of each electrode based on histology. All results are expressed as Fleiss’ kappa, with a target alpha of 0.05.

		All Electrodes	Scala Tympani Electrode	Intermediate Electrode	Scala Vestibuli Electrode
**Advanced Bionics MidScala**	Expert	0.76	0.79	0.24	0.89
	Non-expert	0.67	0.75	0.06	0.81
	Expert + Non-expert	0.68	0.76	0.12	0.82
**Oticon EVO**	Expert	0.46	0.55	0.24	0.60
	Non-expert	0.39	0.51	0.13	0.41
	Expert + Non-expert	0.39	0.51	0.16	0.42

## Data Availability

The data presented in this study are available on request from the corresponding author.

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
