# Peer review of "Best Fit 3D Basilar Membrane Reconstruction to Routinely Assess the Scalar Position of the Electrode Array after Cochlear Implantation"

_jcm, 2022, doi:10.3390/jcm11082075_

Round 1
Reviewer 1 Report
Figure 1. H360 was measured from the base of the cochlea to the highest part of the cochlear duct at 360° (white arrow). This is not cochlear duct. Maybe the authors mean one cochlear turn (highest part of the cochlear turn at 360 degrees. This should also be mentioned under figure c. I believe the authors intermix cochlear duct and cochlear turn. Cochlear duct is the membraneous part.
Author Response
We thank you for your suggestions and comments.
Comments: Figure 1. H360 was measured from the base of the cochlea to the highest part of the cochlear duct at 360° (white arrow). This is not cochlear duct. Maybe the authors mean one cochlear turn (highest part of the cochlear turn at 360 degrees. This should also be mentioned under figure c. I believe the authors intermix cochlear duct and cochlear turn. Cochlear duct is the membraneous part.Response: I agree with you that the correct term is cochlear "turn" instead of cochlear "duct". This term has been changed throughout the document.
Reviewer 2 Report
Results are now good presented
Author Response
We thank you for your comments and suggestions.
This manuscript is a resubmission of an earlier submission. The following is a list of the peer review reports and author responses from that submission.
Round 1
Reviewer 1 Report
This is an interesting study with a generally well written and structured methodology description. Figure captions are clear enough and the procedure is carefully described.
However, in my opinion, the introduction should identify the subject area of interest in more detailed and the state of art should be examined in more depth.
Above all, I suggest to make the study as complete as possible, even though the paper is presented as a “stage one”. There is a total lack of a comments and a result and discussion section might be useful and could help to keep the manuscript in a more interesting and in-depth format. And, besides, the abstract includes information which are not mentioned in the article full text (for example: the time of analysis, Fleiss' kappa statistical measure, authors' considerations).
Author Response
We would like to thank you for your comments and critics. We hope that the modifications will fulfill your questions and improve the understanding of the manuscript.
Comment: The introduction should identify the subject area of interest in more detailed and the state of art should be examined in more depth.
Response: The introduction has been substantially modified. The different methods to analyze the electrode array position have been specified, the advantages and limitation of each of them specified and the goals of the method we propose stated.
Comments: Above all, I suggest to make the study as complete as possible, even though the paper is presented as a “stage one”. There is a total lack of a comments and a result and discussion section might be useful and could help to keep the manuscript in a more interesting and in-depth format. And, besides, the abstract includes information which are not mentioned in the article full text (for example: the time of analysis, Fleiss' kappa statistical measure, authors' considerations).
Response: As the first submission was a stage 1 manuscript, only the introduction and material & methods were included. Now, the final version of the article has been submitted including results and discussion sections.
Reviewer 2 Report
This paper is about a new clinical method to assess the position of the CI electrode relative the basilar membrane following implantation to verify scalar dislocation. The authors establish electrode position from clinical CT and compares it with micro-CT of temporal bone specimens as reference. A collection of ten basilar membrane templates was generated from micro-CT, based on the dimensions of 100 cochleae. Five ENT surgeons blindly determined the position of the electrode array of 30 cadaveric cochleae. Data suggest that using an adapted basilar membrane template can be used as a rapid and reproducible method to assess the position of the electrode array after cochlear implantation. This may be used to compare new innovative strategies to insert electrodes such as via robot. Hopefully, such data can lead to avoidance of future dislocations at surgery that could influence functional outcome.
The procedure consists in “selecting the appropriate template based on cochlea dimensions, merging the electrode array reconstruction with the template using four landmarks, determining the position of the array according to the template position, and comparing the obtained results to histology.”
The authors define the middle line of the cochlear turn as basilar membrane. The membrane is believed to extend from the medial cochlear wall to the lateral wall including the spiral lamina. It is unfortunate but may be more a matter of definition. The human basilar membrane is different in width in the basal compared to the apical turn. The ”basilar membrane” here is rather a middle plane of the cochlear turn on which the center of the basilar membrane is partly lodged. This must be clearly stated.
There are several concerns about the measurements of distances. A coronal plane is not good since the human cochlea is very differently placed in the skull among individuals. Better is to use Stenver’s projection that aligns the petrous pyramid so that the cochlea is best projected “en face” with the center of the round window and the lateral wall around 360 degrees.
Figure 1b does not seem to be a sagittal view. It seems to be a horizontal view. The measurement of H360 seems incorrect. It is not 360 degrees which is one turn. This distance does not either represent the highest part of the cochlear duct nor 360 degrees. The cochlear duct continues 950 degrees to the apical third turn of the cochlea. Text in Figure 1d is too small to be readable.
In figure 2, a coronal plane is used. Why not use Stenver’s projection to be sure that the cochlea is properly aligned? A coronal section gives different results in different individuals due to the great variations of the cochlea in the skull. How can the coronal plane be aligned to the middle of the cochlear duct?
Figure 3. This is not a good image stating that it is a segmentation of the basilar membrane. This is not the basilar membrane. It is the middle plane of the cochlear turn. "Basilar" in the first sentence should be "basilar membrane" in figure legend. The authors state that it represents the basilar membrane but it also includes the spiral lamina. Basilar membrane looks different in humans having different sizes at different frequency locations.
Figure 5. The histological image is difficult to understand. Is this a microscopic section, H&E stained section? Why is the histologic technique not mentioned in M&Ms.
Pertinent literature is given but there is a lack of literature on human basilar membrane anatomy.
Author Response
We would like to thank you for your comments and critics. We hope that the modifications will fulfill your questions and improve the understanding of the manuscript.
This paper is about a new clinical method to assess the position of the CI electrode relative the basilar membrane following implantation to verify scalar dislocation. The authors establish electrode position from clinical CT and compares it with micro-CT of temporal bone specimens as reference. A collection of ten basilar membrane templates was generated from micro-CT, based on the dimensions of 100 cochleae. Five ENT surgeons blindly determined the position of the electrode array of 30 cadaveric cochleae. Data suggest that using an adapted basilar membrane template can be used as a rapid and reproducible method to assess the position of the electrode array after cochlear implantation. This may be used to compare new innovative strategies to insert electrodes such as via robot. Hopefully, such data can lead to avoidance of future dislocations at surgery that could influence functional outcome.
The procedure consists in “selecting the appropriate template based on cochlea dimensions, merging the electrode array reconstruction with the template using four landmarks, determining the position of the array according to the template position, and comparing the obtained results to histology.”
Comments: The authors define the middle line of the cochlear turn as basilar membrane. The membrane is believed to extend from the medial cochlear wall to the lateral wall including the spiral lamina. It is unfortunate but may be more a matter of definition. The human basilar membrane is different in width in the basal compared to the apical turn. The “basilar membrane” here is rather a middle plane of the cochlear turn on which the center of the basilar membrane is partly lodged. This must be clearly stated.
Response: We agree with your remark concerning the variation in length of the basilar membrane in the basal and apical part of the cochlea. We segmented the division of the scala tympani and the scala vestibuli, which included the bony spiral lamina and the basilar membrane.
Now, the middle plane of the cochlear duct will be well defined on the M&M section (lines 156-159) and it will be specified that the middle plane of the cochlear duct will be called “basilar membrane” in quotes throughout of the article.
Comments: There are several concerns about the measurements of distances. A coronal plane is not good since the human cochlea is very differently placed in the skull among individuals. Better is to use Stenver’s projection that aligns the petrous pyramid so that the cochlea is best projected “en face” with the center of the round window and the lateral wall around 360 degrees.
Response: I agree with you that the orientation of the cochlea in the temporal bone is different among people. As we worked in a tridimensional space, and used multiplanar reconstructions, the intersection of the three planes was placed at the level of the mid-modiolar axis, the coronal plane was aligned with the middle plane of the basal turn (center of the round window, the middle of the cochlear duct at 90, 180 and 270 degrees). Consequently, the axial plane will be perpendicular to the coronal plane and crossed the center of the round window and the cochlear duct at 180°, and the sagittal plane will be also perpendicular to the coronal plane crossing the cochlear duct at 90° and 270°.
Line 79-82: original version
The cochlea was aligned as follows: 1) in the coronal plane, the intersection of the three planes were placed on the modiolus and the axial plane aligned with the center of the round window (RW) and the lateral wall at 180°, 2) the sagittal plane aligned with the lateral wall of the cochlea at 90° and 270°.
Line 110-116: last version
The cochlea was aligned as follows: 1) the intersection of the three planes was placed on the mid-modiolar axis; 2) the coronal plane was aligned with the middle plane of the basal turn (center of the round window, the middle of the cochlear duct at 90°, 180°, and 270°); 3) the axial plane was consequently perpendicular to the coronal plane and passed through the center of the round window and the cochlear duct at 180°; and 4) the sagittal plane was also perpendicular to the coronal plane and passed through the cochlear duct at 90° and 270°.
Comments: Figure 1b does not seem to be a sagittal view. It seems to be a horizontal view. The measurement of H360 seems incorrect. It is not 360 degrees which is one turn. This distance does not either represent the highest part of the cochlear duct nor 360 degrees. The cochlear duct continues 950 degrees to the apical third turn of the cochlea. Text in Figure 1d is too small to be readable.
Response: The figure 1b is the sagittal view, in this case the figure was rotated to have the basal turn inferiorly and the apical turn superiorly. Now, the fact that this figure was rotated and the modification to the figure has been stated on the figure legend.
Line 90: original version
(b) sagittal view
Lines 125-126: last version
(b) sagittal view, which has been rotated to position the basal turn inferiorly for better visualization,
In this figure, the height at 360° represents the distance between the base of the cochlea to the highest point of the cochlear duct at 360°. If we observe the figure 1a (coronal view) the center of the round window is 0 degrees and 360 degrees correspond to the first complete turn. Figure 1c correspond to the axial view, which is aligned with the center of the round window and the cochlear duct at 180°. At this level, the we could see the first complete turn of the cochlea (360°). The height of the cochlea at this point (white arrow) was measured from the base of the cochlea to the parallel plane passing through this highest point of the cochlear duct at this level. We don’t considerate the full cochlear height as measure as stated in the M&M section.
The figure was modified in order to improve the quality of the figure, there were added the position in degrees in all planes and remove the text in figure 1d which was unreadable and confusing.
Original figure (lines 87-88)
Last version of the figure (lines 121-122)
Comments: In figure 2, a coronal plane is used. Why not use Stenver’s projection to be sure that the cochlea is properly aligned? A coronal section gives different results in different individuals due to the great variations of the cochlea in the skull. How can the coronal plane be aligned to the middle of the cochlear duct?
Response: The cochlear orientation according to the temporal bone structures is variable. In consequence, the coronal view was aligned with the middle plane of the basal turn (center of the round window, middle of the cochlear duct at 90, 180 and 270). This alignment allowed to align the coronal plane with the middle plane of the basal turn, regardless the orientation of the cochlea in the temporal bone. So, we aligned the three planes according to the orientation of the cochlea regardless the orientation of the cochlea in the temporal bone or the misalignment of the head during the acquisition of the CT, and this is easily achieved thanks to the multiplanar reconstruction.
Comments: Figure 4. This is not a good image stating that it is a segmentation of the basilar membrane. This is not the basilar membrane. It is the middle plane of the cochlear turn. "Basilar" in the first sentence should be "basilar membrane" in figure legend. The authors state that it represents the basilar membrane but it also includes the spiral lamina. Basilar membrane looks different in humans having different sizes at different frequency locations.
Response: I agree with you that the basilar membrane length in variable along the cochlea and the 3D reconstruction represents the middle plane of the cochlea that includes the basilar membrane and the spiral lamina. Consequently, the middle plane of the cochlear duct will be well defined on the M&M section (lines 156-159) and it will be specified that the middle plane of the cochlear duct will be called “basilar membrane” in quotes throughout of the article.
Comments: Figure 5. The histological image is difficult to understand. Is this a microscopic section, H&E stained section? Why is the histologic technique not mentioned in M&Ms.
Response: I agree with you that the histological analysis was not mentioned in the M&M section. Now, the histological analysis was described as follows, and the histology section of the figure modified for better understanding.
Lines 269-280
- Description of the histopatological analysis
2.6. Description of the histopathological analysis
All cochleae used in this study were analyzed and the results stored in our database. After electrode array insertion, the cochlea was harvested, and the apex of the cochlea and lateral semicircular canal were opened. The cochlea was then fixed with 10% formaldehyde for 24 hours, dehydrated with increasing concentrations of alcohol (from 50% to 100%) for 12 hours, dried at ambient air temperature for 16 hours and fixed with a crystal resin (Pebeo, Gémenos, France) until polymerization, as reported in a previous study [20]. The cochlea was then progressively ground perpendicular to the round window/modiolar axis and stopped at the level of each electrode. The cochlea was stained with Phloxine B for 15 minutes and the cochlea was visualized with a stereomicroscope (SLM 2; Karl Kaps GmbH, Wetzlar, Germany) and a photograph was taken and stored in our database.
In addition, the figure has been modified to delimit the cochlear duct and point at the basilar membrane.
Original figure: 212-217
Figure 5. Basilar membrane reconstruction (a) and histologic (b) techniques to assess the position of each electrode array. The basilar membrane reconstruction used was selected based on the dimensions of the analyzed cochlea. ST: scala tympani, SV: scala vestibuli, black arrow: electrode array
Modified figure and legend: 260-268
Figure 5. The middle plane of the cochlear duct reconstruction and histologic techniques were used to assess the position of each electrode array. (a) The “basilar membrane” reconstruction, including the basilar membrane and spiral lamina, was selected based on the dimensions of the cochlea analyzed. (b) The microgrinding technique shows the electrode array penetrating the scala tympani. The black discontinuous lines delimit the cochlear duct. ST: scala tympani; SV: scala vestibuli; black arrow: electrode array; blue arrow: basilar membrane.
Comments: Pertinent literature is given but there is a lack of literature on human basilar membrane anatomy.
Response: We agree with you, references were added concerning the variability of the cochlear anatomy in the introduction, and the discussion.
Lines 457-462
- Biedron, S.; Prescher, A.; Ilgner, J.; Westhofen, M. The Internal Dimensions of the Cochlear Scalae with Special Reference to Cochlear Electrode Insertion Trauma. Otol Neurotol 2010, 31, 731–737, doi:10.1097/MAO.0b013e3181d27b5e.
- Escudé, B.; James, C.; Deguine, O.; Cochard, N.; Eter, E.; Fraysse, B. The Size of the Cochlea and Predictions of Insertion Depth Angles for Cochlear Implant Electrodes. Audiol Neurootol 2006, 11 Suppl 1, 27–33, doi:10.1159/000095611.
- Erixon, E.; Högstorp, H.; Wadin, K.; Rask-Andersen, H. Variational Anatomy of the Human Cochlea: Implications for Cochlear Implantation. Otol Neurotol 2009, 30, 14–22, doi:10.1097/MAO.0b013e31818a08e8.

Reviewer 3 Report
Data like 12 min per Cochlea or all Kappas from the abstract are not reported in the main text. Reading the manuscript it feels like ending at an interesting point. What are the results of the expert versus untrained users? Show the comparison, show the results! The results and discussion are missing.
Some smaller aspects:
Line 24: Please reword: “analysis was about 12 min for cochlea”
Line 53: “This is particularly true…” … what do you mean?
Line 63: Please clarify in the introduction what you have done. It is not that clear until here.
Line 76: post OP Scans?
Line 133: How do you know that 5 templates are enough?
Line 153-155: Please reword; in general please try to clarify what is your work and what did you take from a database
Line 155: Please change the wording … “a selected the best fitted”
Line 184: Please change the wording… “to under”
Line 198. Better “in handling 3D reconstruction…”
Figure 1: Caption: The white continuous arrow is mentioned twice.
Figure 3: Why colorbar not just a square?
Author Response
We would like to thank you for your comments and critics. We hope that the modifications will fulfill your questions and improve the understanding of the manuscript.
Comment: Data like 12 min per Cochlea or all Kappas from the abstract are not reported in the main text. Reading the manuscript, it feels like ending at an interesting point. What are the results of the expert versus untrained users? Show the comparison, show the results! The results and discussion are missing.
Response: Because of the manuscript was a stage 1 review, we sent the introduction and M&M, now, the version submitted includes the results and the discussion.
Some smaller aspects:
Line 24: Please reword: “analysis was about 12 min for cochlea”
Response: the phrase was modified as follows: . The time taken to analyze each implanted cochlea was approximately 12 minutes. (line 23-24)
Line 53: “This is particularly true…” … what do you mean?
Response: Because the introduction has been substantially changed, this phrase was removed from the manuscript.
Line 63: Please clarify in the introduction what you have done. It is not that clear until here.
Response: We agree with you, consequently, the introduction has been substantially modified in order to better identify the area of interest.
Line 76: post OP Scans?
Response: The measures were obtained in 100 preoperative CT scan of patients, 22 non-implanted micro CT and the 30 preoperative CBCT of the cadaveric temporal bones. Consequently, the word preoperative CT has been added in the manuscript (line 107).
Line 133: How do you know that 5 templates are enough?
Response: In a preliminary study, we analyzed cochleae using first one template, then three templates and five templates. Using only one template to determine the position of the electrode was insufficient, however there were no difference in determine the scalar position using 3 templates compared to 5 templates. Consequently, we have considered that increase the number of templates to 7 will not change the results.
Line 153-155: Please reword; in general, please try to clarify what is your work and what did you take from a database
Response: We segmented all the basilar membranes from CBCT images of cadaveric temporal bones. I agree with you that the word “bank” is confusing, so this phrase has been modified as follows.
Original text (lines 153-155): 10 CBCT images obtained from cadaveric temporal bones were selected from our image bank, from which we obtained the basilar membrane reconstruction and four landmarks.
Modified text (lines 192-193): we reconstructed 10 “basilar membranes” from CBCT images of cadaveric temporal bones.
Line 155: Please change the wording … “a selected the best fitted”
Response: the phrase “selected the best fitted basilar membrane” has been modified as follow: … and the corresponding “basilar membrane” template was selected. (Line 193-194)
Line 184: Please change the wording… “to under”
The original statement: Intermediate electrode: ≥10% to <50% of the electrode to under the basilar membrane
It was modified as follow: Intermediate electrode: ≥10% to <50% of the electrode under the basilar membrane (line 229-230)
Line 198. Better “in handling 3D reconstruction…”
Response: The original statement: Two ENT surgeons were considered as experts due to their experience in handle 3D reconstruction models.
It was modified as follow: (Line 243-245) Two ENT surgeons were considered experts because of their experience in handling 3D reconstruction models.
Figure 1: Caption: The white continuous arrow is mentioned twice.
Response: We agree with your remark, the statement “White arrow: top of the cochlear duct at 360°” was removed from the legend.
Figure 3: Why colorbar not just a square?
Response: I agree with you that the colorbar is not regular. The figure has been modified as follows
Original figure
Modified figure with legend (line 208-218)
Figure 4. Comparison of basilar reconstruction segmented from micro-CT images and CBCT. The term basilar membrane represents the middle plane of the cochlear duct which includes the basilar membrane and the spiral lamina. (a) The hook region has not been segmented on CBCT images because of its complexity (open square). (b) The distance between both segmented “basilar membranes” (both cropped from 90° to 540°) was calculated. The color scale represents the distance between both basilar membranes, from green (shorter gap) to red (greater gap). In grey, both basilar membranes are superposed.

Reviewer 4 Report
In this interesting studies authors aimed to present a method for determining the postoperative scalar CI electrode position. Furthermore, the assessed the observer intra-variability. They analyzed and tested it in 100 CT images of patients later becoming a CI, 30 cadaver cochleae and 22 non implanted ears.
The methods are interesting and are properly presented. It would be necessary that authors go into literature research and provide information of studies already reporting on similar/same measurements.
Furthermore, the clinical utilization remains somewhat unclear. Authors need to discuss their results. Currently, only the methods are thoroughly explained and I fail to see any real results and discussion of these results in the manuscript (at least in the version I am seeing).
Author Response
In this interesting study authors aimed to present a method for determining the postoperative scalar CI electrode position. Furthermore, the assessed the observer intra-variability. They analyzed and tested it in 100 CT images of patients later becoming a CI, 30 cadaver cochleae and 22 non-implanted ears.
The methods are interesting and are properly presented. It would be necessary that authors go into literature research and provide information of studies already reporting on similar/same measurements.
Furthermore, the clinical utilization remains somewhat unclear. Authors need to discuss their results. Currently, only the methods are thoroughly explained and I fail to see any real results and discussion of these results in the manuscript (at least in the version I am seeing).
Response: The first version manuscript (stage 1) included the introduction and M&M. Now, the submitted version includes results and discussion section.